# Automatic Diagnosis of Infectious Keratitis Based on Slit Lamp Images Analysis

**DOI:** 10.3390/jpm13030519

**Published:** 2023-03-13

**Authors:** Shaodan Hu, Yiming Sun, Jinhao Li, Peifang Xu, Mingyu Xu, Yifan Zhou, Yaqi Wang, Shuai Wang, Juan Ye

**Affiliations:** 1Department of Ophthalmology, College of Medicine, The Second Affiliated Hospital of Zhejiang University, Hangzhou 310009, China; 2School of Mechanical, Electrical and Information Engineering, Shandong University, Weihai 264209, China; 3College of Media Engineering, Communication University of Zhejiang, Hangzhou 310018, China; wangyaqi@cuz.edu.cn; 4Suzhou Research Institute, Shandong University, Suzhou 215123, China

**Keywords:** deep learning, infectious keratitis, slit lamp image, automatic classification

## Abstract

Infectious keratitis (IK) is a common ophthalmic emergency that requires prompt and accurate treatment. This study aimed to propose a deep learning (DL) system based on slit lamp images to automatically screen and diagnose infectious keratitis. This study established a dataset of 2757 slit lamp images from 744 patients, including normal cornea, viral keratitis (VK), fungal keratitis (FK), and bacterial keratitis (BK). Six different DL algorithms were developed and evaluated for the classification of infectious keratitis. Among all the models, the EffecientNetV2-M showed the best classification ability, with an accuracy of 0.735, a recall of 0.680, and a specificity of 0.904, which was also superior to two ophthalmologists. The area under the receiver operating characteristics curve (AUC) of the EffecientNetV2-M was 0.85; correspondingly, 1.00 for normal cornea, 0.87 for VK, 0.87 for FK, and 0.64 for BK. The findings suggested that the proposed DL system could perform well in the classification of normal corneas and different types of infectious keratitis, based on slit lamp images. This study proves the potential of the DL model to help ophthalmologists to identify infectious keratitis and improve the accuracy and efficiency of diagnosis.

## 1. Introduction

Corneal opacity is the fifth-leading cause of blindness worldwide [1], and infectious keratitis (IK) is the leading cause of corneal blindness in both developed and developing countries [2]. The most prominent feature of IK is that the growth of pathogens in the cornea leads to local opacity and roughness, and each pathogen shows its unique characteristics in the growth [3]. According to the types of pathogens, infectious keratitis can be divided into bacterial keratitis (BK) [4], fungal keratitis (FK) [5], and viral keratitis (VK) [6]. Once corneal infection occurs, it may progress rapidly, leading to irreversible visual impairments such as corneal scars, endophthalmitis, and corneal perforation [7]. Therefore, early detection and timely medical intervention are critical to stop or slow the progression of the infection.

Corneal scrape culture is currently the gold standard for the diagnosis of IK, but there are drawbacks, such as the risk of corneal injury, the low positive rate of culture, and long diagnostic cycles [8,9]. New techniques such as polymerase chain reaction (PCR) [10,11,12] and confocal microscopy [13] have also been used clinically to assist in diagnosis, but these methods require sophisticated equipment, complex procedures and experienced technicians. At present, the initial diagnosis of IK is highly dependent on ophthalmologists, who need to combine personal experience to distinguish the visual features of corneal lesions. Slit lamp microscopy is the most common ophthalmologic examination used to evaluate the appearance of IK. Slit lamp photographs are commonly used to record and monitor the progression of IK [14]. However, because of the diversity of pathogens and the similarity of lesion manifestations, it is difficult to identify different IK, even for experienced corneal specialists. Overall, there is still a lack of an efficient and accurate diagnostic tool to help guide the treatment of infectious keratitis in clinical practice.

In recent years, artificial intelligence (AI) technology has developed rapidly, and medicine has become the frontier area of AI applications. Recent studies of deep learning (DL) have shown great promise in the use of clinical images to detect common diseases [15]. The convolutional neural network (CNN), as its representative algorithm, has been proven to be very effective in medical image recognition and classification [16]. The automatic classification of medical images can not only reduce the workload of doctors and improve the efficiency and repeatability of screening procedures, but also improve patient outcomes through early detection and treatment. Although there are more than 200,000 ophthalmologists worldwide, there is a current and anticipated future shortage in the number of ophthalmologists in both developing and developed countries [17]. The widening gap between demand and supply can affect the timely detection of infectious keratitis, especially in remote or medically underserved areas [18]. In situations where ophthalmologists are in short supply or medical resources are limited, artificial intelligence is expected to become a practical tool for front-line medical care. 

In the field of ophthalmology, a large number of studies have developed high-precision AI diagnostic systems based on rich examinations and image data for diseases of the posterior segment of the eye, such as diabetic retinopathy, glaucomatous optic neuropathy, and retinal detachment [19,20,21]. The application progress of deep learning in different fundus diseases has extended to early screening, grading and stage diagnosis and even to the prediction of treatment effects [22,23,24]. In contrast, the application of deep learning in anterior segment diseases is limited and has great research potential. Recent studies have attempted to apply deep learning to slit lamp images for the diagnosis of corneal diseases. Gu et al. [25] developed a DL model to identify infectious keratitis, non-infectious keratitis, corneal dystrophy or degeneration, and corneal neoplasm, with results similar to their ophthalmologists. However, to our knowledge, the application of DL to the classification of pathogens in infectious keratitis remains limited.

Thus, we decided to construct a deep learning system for the automatic classification of slit lamp images to achieve an intelligent diagnosis of infectious keratitis, which could offer assistance to ophthalmologists in screening and diagnosing infectious keratitis. We hope that it will help reduce the rate of clinical misdiagnoses, save patients’ vision, and further alleviate the burden on medical resources and society’s economy.

## 2. Methods

### 2.1. Image Dataset 

This study retrospectively established a dataset that included 2757 slit lamp images collected from 744 patients between August 2016 and September 2021 in the Eye Center at the Second Affiliated Hospital of the Zhejiang University School of Medicine. The images were acquired by experienced ophthalmic technicians using two Topcon SL-D701 slit lamp biomicroscopes affixed with DC-4 digital cameras in the diffuse illumination mode. In the slit lamp image dataset, 2165 images were taken from IK patients at the active stage, including bacterial keratitis (BK), fungal keratitis (FK) and viral keratitis (VK). In addition, 592 images taken from healthy eyes with negative fluorescence staining were classified into the category of normal cornea. The representative images for each category are shown in Figure 1.

All cases of IK were diagnosed by cornea specialists from our center, based on medical histories, clinical manifestations, corneal examinations, laboratory methods, and follow-up outcomes. The diagnostic labels based on the medical records’ information were considered to be the true labels of this study. The diagnostic criteria were (1) BK: positive corneal scraping results for bacteria (microscopic staining or tissue culture); (2) FK: positive corneal scraping results for fungi (microscopic staining or tissue culture) or fungal hyphae found under confocal microscopy; (3) VK: positive PCR test results for viruses on corneal scraping; (4) the corresponding typical clinical history; (5) the corresponding typical manifestations of corneal lesions; (6) the corresponding anti-pathogenic drugs were effective. The first three are laboratory gold standards, wherein one of which must be met; the last three are auxiliary indicators, at least one of which must be met. The exclusion criteria were (1) patients with not enough evidence for a definite diagnosis or with mixed infections; (2) patients with corneal perforation, corneal scarring, a history of corneal surgery, or other corneal diseases; (3) images with poor quality, including poor-field, defocused, and poor-location images. Images that met any of the above criteria were excluded. Our two researchers independently reviewed all data in detail before any analysis and ensured that each image was correctly matched to the specific individual.

### 2.2. Data Preparation

The slit lamp images were initially preprocessed, including with shuffle and normalization. For each category of images, the dataset was divided into a training set, validation set and test set. For reducing the impact of the dataset imbalance, the ratio of the BK was adjusted to 0.65:0.2:0.15 and the other three groups were 7:1:2. All images collected from the same patient were only divided into the same dataset to avoid data information leakage from the test set and incorrect evaluation of the model performance. The distribution of the slit lamp image datasets is shown in Table 1.

Data augmentation is an essential approach to automatically generate new training samples and improve the generalization of the DL models [26]. We obtained samples using several strategies: (1) random cropping images with ranges of (0%, 30%); (2) random rotation with ranges of (−10°, +10°); (3) color jitter with ranges of (−10%, + 10%); (4) random horizontal flip; (5) adding Gaussian noise with mean = 0, variance = 1, and amplitude = 8. In addition, we tripled the sample size of the BK to reduce the effect of dataset imbalance. Finally, each image was resized to 224 × 224 pixels to be compatible with the original dimensions of the experiment networks.

### 2.3. Deep Learning Model

Figure 2A shows the flowchart of the DL system for the automatic diagnosis of IK based on slit lamp images. For the development of the DL diagnosis models, we applied various classical and efficient algorithms, including VGG16 [27], ResNet34 [28], InceptionV4 [29], DenseNet121 [30], ViT-Base [31] and EffecientNetV2-M [32]. The above networks are all CNN structures, except for ViT, which uses the Transformer structure. As the latest algorithm proposed in 2021, EffecientNetV2 has achieved state-of-the-art performance in many major image classification tasks, and its structure is shown in Figure 2B.

The training set was used to train the DL models for differentiating infectious keratitis, whereas the validation set was used to verify their performance. All models were implemented using the Pytorch platform with an Nvidia RTX 2080TI GPU. For each model, 150 epochs were set for the training, and the batch size was set at 15. During model training, RMSPromp optimizer and cosine annealing were applied to help the model converge quickly. Moreover, hyperparameter tuning was used to optimize all models according to the validation results. The test set was used to evaluate the performance of the optimal model. The heatmaps were plotted by the Gradient-weighted Class Activation Mapping (Grad-CAM) technique [33]. It can generate visual interpretations for CNN-based deep learning models to build trust in the predicted results and provide references for doctors.

### 2.4. Performance Assessment

Six different DL models in this study were evaluated in an independent test set, and the images obtained from the same patient were not scattered to different datasets. The performance of the DL models for the classification of IK was evaluated by calculating the accuracy, recall and specificity. Statistical analyses were conducted using Python 3.8.10 (Wilmington, DE, USA) and Pycharm 2021.1.3 (Professional edition). To compare the classification ability of the models, the receiver operating characteristics (ROC) curves were created using the packages Scikit-learn (version 0.24.2) and Matplotlib (version 3.2.2). The horizontal axis of the ROC curve is the false positive rate (FPR), which is 1-specificity, and the vertical axis is the true positive rate (TPR), which is the recall. The area under the ROC curve (AUC) can measure classification performance. The closer the value of AUC is to 1, the better the performance is. We also recruited two ophthalmologists to independently classify the same test set and evaluate their classification results using the same metrics. Then, the classification performance of the best-performing model was compared by two ophthalmologists (i.e., Doctor 1 and Doctor 2). The confusion matrices were plotted by Matplotlib (version 3.2.2), which is helpful in analyzing the misclassification of each category by the model and ophthalmologists.

## 3. Results

### 3.1. Performance of the DL Models

Six DL algorithms were used in this study to train the models for the classification of BK, VK, FK and normal cornea. The performance of these DL models in the test set was evaluated by accuracy, recall, and specificity, as shown in Table 2. The best classification algorithm was EffecientNetV2-M, with an accuracy of 0.735, a recall of 0.680, and a specificity of 0.904. Doctor 1 and Doctor 2 reached an accuracy of 0.661 and 0.685, a recall of 0.636 and 0.648, and a specificity of 0.884 and 0.891, respectively.

The macro-average ROC curves of the six DL models are shown in Figure 3. InceptionV4 reached the highest AUC value of 0.86, while EffectNetV2-M reached 0.85, higher than other models (VGG16 AUC = 0.83, ResNet34 AUC = 0.82, ViT-Base AUC = 0.82, DenseNet121 AUC = 0.81).

### 3.2. Comparison with the Ophthalmologists

Figure 4A compares the macro-average ROC curves between the EffecientNetV2-M model and the ophthalmologists. The EffecientNetV2-M model achieved an AUC of 0.85, higher than two ophthalmologists, which were 0.76 and 0.77, respectively. The ROC curves of the EffecientNetV2-M model for each category are shown in Figure 4B, with Doctor 1 corresponding to Figure 4C and Doctor 2 corresponding to Figure 4D. For the AUCs of each category, the EffecientNetV2-M model, Doctor 1 and Doctor 2 reached a normal cornea AUC of 1.00, 0.89 and 0.97, respectively; VK of 0.87, 0.75 and 0.78, respectively; FK of 0.87, 0.74 and 0.72, respectively; and BK of 0.64, 0.66 and 0.61, respectively.

Figure 5A,B shows the confusion matrices of the EffecientNetV2-M model and the average results of the two ophthalmologists. The horizontal axis represents the true category labels, and the vertical axis represents the predicted category labels. As the green color of the matrix deepens, it means that the value increases. The distribution of the last line shows the difficulty of classifying BK, where the real BK is easily misjudged as FK, either by the DL model or by the ophthalmologists. In addition, it can be seen that the model is more likely to recognize the real VK as the other groups, whereas the ophthalmologist is more likely to recognize the other groups as VK. The confusion matrix reveals the similarities and differences between the algorithm and the ophthalmologists in misclassification.

### 3.3. Heatmaps

Figure 6 presents examples of heatmaps generated by the EffecientNetV2-M model, accompanied by the corresponding original image. The redder regions represent the areas that are of greater concern during the model classification process, and the bluer regions represent those of relatively less concern. The heatmaps highlighted the areas with corneal lesions, which are highly correlated with the identification of infectious keratitis.

### 3.4. Discussion

As a common emergency in ophthalmology, timely and accurate treatment is essential for the prognosis of infectious keratitis. However, the diagnosis of infectious keratitis is a huge challenge clinically due to the diversity of pathogens and the similarity of clinical manifestations. In this study, we constructed a DL diagnosis system for IK to automatically classify BK, FK, VK, and normal corneas by analyzing slit lamp images. Six DL models were developed using the same dataset and compared with the performance of two ophthalmologists. In the end, the EffecientNetV2-M model performed better than the other models and the ophthalmologists, with an accuracy of 0.735, a recall of 0.680, and a specificity of 0.904. Our results suggested that the DL model could be useful for ophthalmologists to screen and diagnose infectious keratitis, thereby reducing the rate of misdiagnosis clinically, saving patients’ vision, and further alleviating the burden of medical resources and socioeconomic issues. In addition, DL technology makes it possible to provide telemedicine services for the diagnosis of IK in places where timely ophthalmologic assessment cannot be performed, such as in rural areas.

Recently, some studies have attempted to apply deep learning to slit lamp images for the diagnosis of keratitis. Kuo et al. [34] identified fungal and nonfungal keratitis using the DenseNet network based on 288 corneal photographs, with an accuracy of 69.4% and an AUC of 0.65. Ghosh et al. [35] adopted the ensemble technique for identifying BK and FK based on 194 cases, obtaining a recall rate of 0.77 and an F1 score of 0.83. Li et al. [36] developed a DL system that can automatically classify keratitis, other cornea abnormalities, and normal corneas with slit lamp and smartphone images, with an AUC of more than 0.96. Xu et al. [37] compared three image-level algorithms for the classification of BK, FK, HSK, and other corneal disorders, with the DenseNet model achieving optimal accuracy (64.17%), which was superior to 421 ophthalmologists (49.27 ± 11.5%). In addition, a large international study [38] quantified the performance of 66 cornea specialists in the image-based differentiation of BK and FK, with AUCs of 0.39–0.82 (mean of 0.61).

Compared with the previous studies, the advantages of this study are as follows. First, we collected and built a relatively large and diverse image dataset, including 2757 slit lamp images from 744 patients. Furthermore, the classification task of this study was relatively more complex—to distinguish BK, FK, VK, and normal cornea. Next, we evaluated and compared the classification performance of six different models and two ophthalmologists. The EffecientNetV2 model outperformed other classical models and ophthalmologists in this study, achieving an AUC of 0.85 and an accuracy of 0.735. The ophthalmologists reached an AUC of 0.76 and 0.77, with an accuracy of 0.661 and 0.685, respectively. Our results were better than most previous studies. It is significant that the EffecientNetV2 algorithm used in this study is currently the latest and strongest CNN, which improves the index by comprehensively optimizing the network width, network depth and resolution. Lastly, visual heatmaps were generated to make our DL system interpretable. The heatmaps of the EffecientNetV2-M model highlighted areas that were highly correlated with the lesions of IK. The interpretability of this model can be useful in real-world applications, as it can help ophthalmologists understand how the DL system produces the final output.

The EffecientNetV2-M model performed best in this study, but its ability to classify different types of IK varied. The AUCs of the normal cornea, VK, FK and BK were 1.00, 0.87, 0.87 and 0.64, respectively. This means that the model has a strong classification ability to distinguish normal cornea images from infectious keratitis and has a good classification effect for VK and FK, but not so good for BK. The possible reasons are as follows: (1) each species of BK may have different lesion characteristics, and the same species may even have different manifestations in different stages of infection, which makes the classification difficult; (2) different types of IK (especially BK and FK) may have similar lesion characteristics, which may lead to their misclassification; (3) the number of images in each group is relatively limited (especially BK), which leads to the failure of the DL models to learn the unique features of various types of keratitis comprehensively. According to the confusion matrices, about half of the real BK images were misjudged as FK by both the DL algorithm and the ophthalmologists. This confirmed that the lesion characteristics of BK and FK are similar, which makes visual diagnosis challenging. In addition, comparing the misclassification between the algorithm and the ophthalmologists may help improve the diagnostic model. However, it must be noted that few clinicians use only a single slit lamp image to diagnose infectious keratitis. The optical quality of the image may affect their judgment, such as in the presence of reflection and artifacts. Therefore, these factors may lead to inconsistency and error when ophthalmologists perform this classification task.

There are also some limitations in this study. First, the dataset was still relatively inadequate compared with some large studies, and cases were not well balanced across groups. In particular, the number of BK images is relatively small, which may also be the main reason for the poor classification effect of BK. The relatively limited number of BK patients may be due to the empirical clinical use of antibiotic eye drops, resulting in a low probability of the laboratory detection of the bacteria. In contrast, bacteria with stronger antibiotic resistance are easier to be detected but are usually accompanied by serious clinical manifestations such as hypopyon. Fortunately, recent studies may have found effective treatments for drug-resistant bacterial corneal infection [39,40]. For us, an early collection of untreated keratitis images may be a solution to increasing the number of BK images. Second, we only evaluated the performance of the developed model on an internal dataset. Various data enhancement processes were used to increase the diversity of the datasets to prevent overfitting and improve the reliability of the training model. However, it is currently difficult to conclude whether the model can be used to screen real patients. In the future, we will collect more cases for model development and validation, including data from multiple centers. Third, we excluded mixed infections with different types of keratitis in this study, although this condition is very common clinically. We believe that identifying mixed infections is a much larger and more difficult deep-learning task. The DL models must fully learn the characterization of each keratitis before it is possible to identify two or more types of keratitis infections. This may require a good understanding of the unique lesion characteristics of different IK, such as corneal infiltration, bacterial or fungal moss, hypopyon and so on. By combining the manual labeling of corneal lesions, it may be possible to improve the focus of the model on the lesion and learn more comprehensive information. In addition, the combination of the model with relevant clinical history information (e.g., trauma, underlying disease, medication history, etc.) may also be a method to enhance model recognition ability. Lastly, our model could not identify the degree or stage of the keratitis infection. We believe that this recognition task needs corneal multi-level information, and we are working on this by combining other corneal examinations, such as fluorescent staining, anterior segment optical coherence tomography (AS-OCT), corneal topography and so on. In any case, we believe that the DL diagnostic system can be further improved to better assist the identification of IK, which can help improve diagnostic accuracy and efficiency.

## 4. Conclusions

In this study, we proposed an intelligent diagnosis system for IK using DL technology to analyze slit lamp images. Compared with the other models and ophthalmologists, the EffecientNetV2-M model achieved higher accuracy and AUC in this study. The findings suggest that the proposed DL system could perform well in the classification of normal corneas and different types of infectious keratitis based on slit lamp images. This demonstrates the potential of the DL model to help ophthalmologists identify infectious keratitis, and it provides the possibility of improving diagnostic accuracy and efficiency.

## Figures and Tables

**Figure 1 jpm-13-00519-f001:**
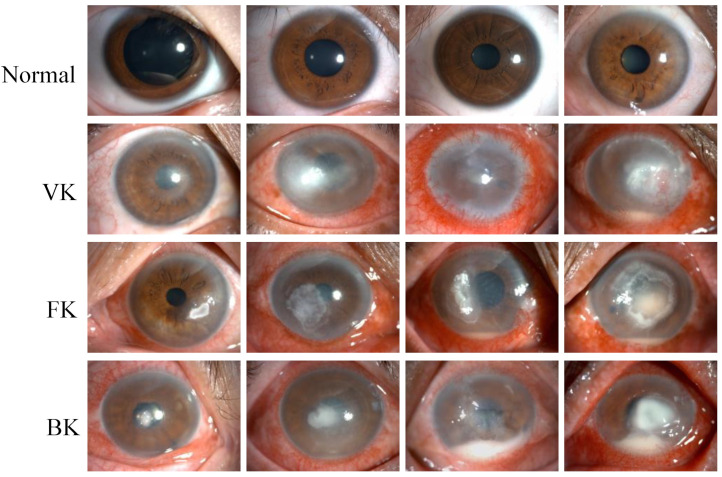
Representative slit lamp images of the normal cornea (Normal), viral keratitis (VK), fungal keratitis (FK), and bacterial keratitis (BK), from top to bottom. Each image was from a different eye, and each category of keratitis showed a different degree of infection.

**Figure 2 jpm-13-00519-f002:**
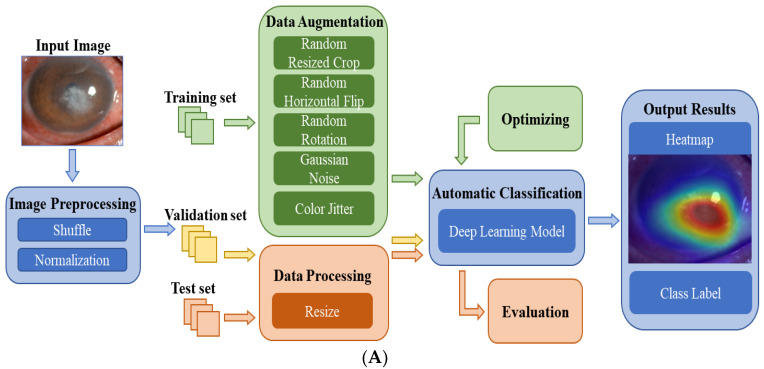
(**A**) The flowchart of the DL diagnostic system for IK based on slit lamp images. Firstly, the input images were preprocessed and divided into three data sets. Next, data augmentation and data processing were performed on the images. Then, various DL algorithms were used for model development, optimization, and evaluation. Finally, the results of the models were output in the form of heatmaps and class labels. (**B**) The architecture of EfficientNetV2. It was mainly composed of several Fused-MBConv blocks and MBConv blocks. Early features were extracted using Fused-MBConv blocks, which used 1 × 1 Conv, 3 × 3 Conv and SE layers. It used depthwise convolution to extract from MBConv blocks. Compared with the traditional convolution widely used in other models, depthwise convolution had fewer parameters and computation.

**Figure 3 jpm-13-00519-f003:**
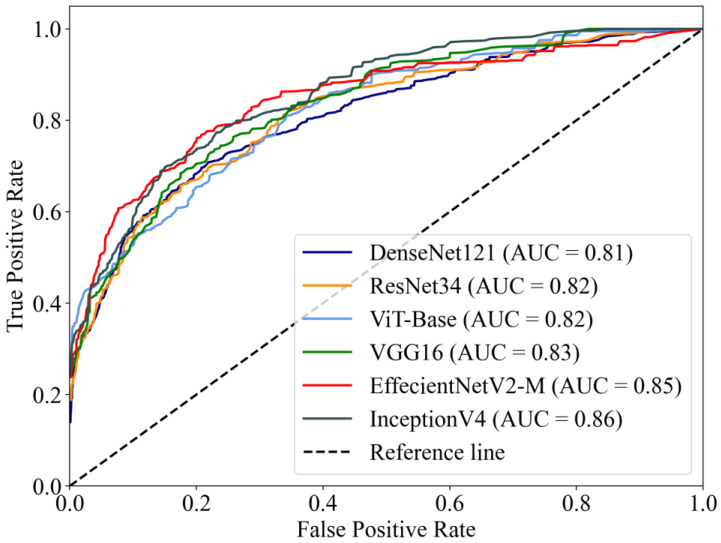
The macro-average ROC curves and AUCs of six DL models. The AUCs of the models ranged from 0.81–0.86.

**Figure 4 jpm-13-00519-f004:**
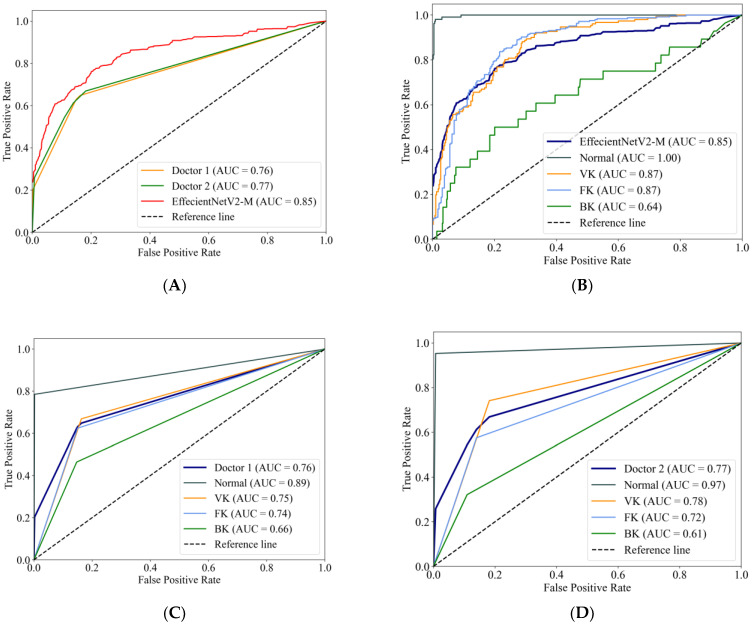
The macro-average ROC curves and AUCs of the EffecientNetV2-M model and two ophthalmologists (**A**). The ROC curves for each category of the EffecientNetV2-M model (**B**), Doctor 1 (**C**) and Doctor 2 (**D**).

**Figure 5 jpm-13-00519-f005:**
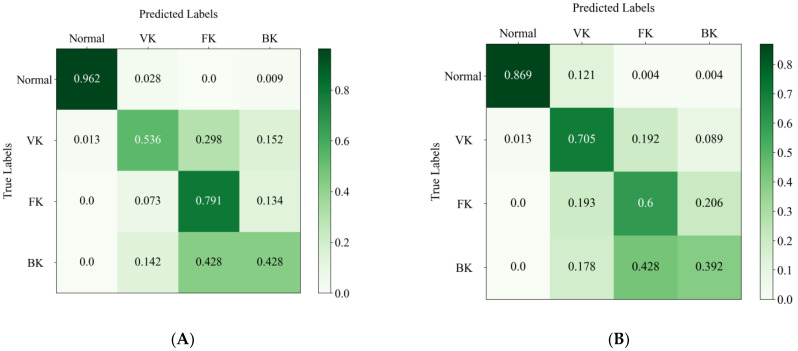
The confusion matrix of the EffecientNetV2-M model (**A**) and the average results of two ophthalmologists (**B**) in the test set. The column denotes the predicted labels, and the row indicates the true labels.

**Figure 6 jpm-13-00519-f006:**
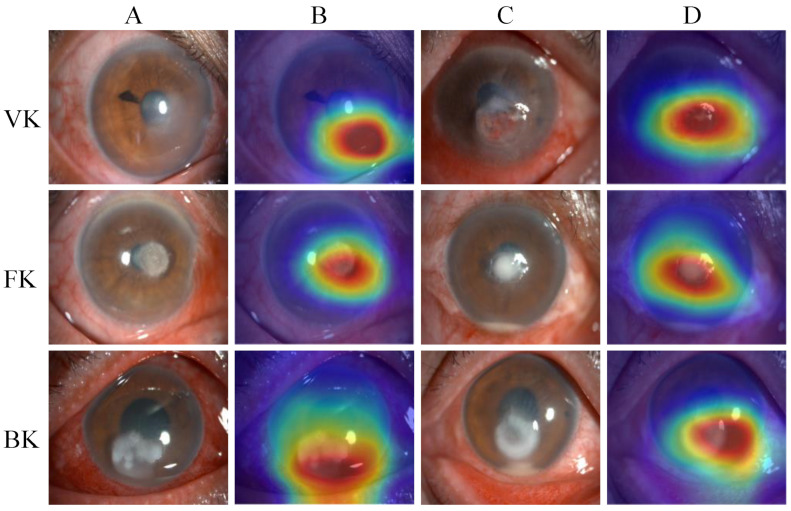
The heatmaps generated by the EffecientNetV2-M model. From top to bottom, each row corresponds to the images of viral keratitis (VK), fungal keratitis (FK) and bacterial keratitis (BK). Columns (**A**,**C**) are different original image examples of images of each category, while column (**B**) are the heatmaps generated by column (**A**), and column (**D**) are the heatmaps generated by column (**C**).

**Table 1 jpm-13-00519-t001:** The distribution of the slit lamp image datasets.

Dataset (*n*)	Normal	VK	FK	BK	Total
Training	425	583	753	164	1925
Validation	60	79	127	35	301
Test	107	151	245	28	531
Total	592	813	1125	227	2757

**Table 2 jpm-13-00519-t002:** Performance comparison of six DL models for IK classification.

Model	Accuracy	Recall	Specificity
ResNet34	0.635	0.554	0.861
DenseNet121	0.637	0.637	0.875
ViT-Base	0.697	0.598	0.888
VGG16	0.708	0.583	0.890
InceptionV4	0.716	0.640	0.897
EffecientNetV2-M	0.735	0.680	0.904

## Data Availability

The data are not publicly available due to privacy issues. Requests to access the datasets should be directed to the Second Affiliated Hospital of Zhejiang University, School of Medicine.

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
