# Peer review of "Automatic Diagnosis of Infectious Keratitis Based on Slit Lamp Images Analysis"

_jpm, 2023, doi:10.3390/jpm13030519_

Round 1
Reviewer 1 Report
The paper describes use of DL (Deep learning) system based on slit lamp images to automatically screen and diagnose infectious keratitis (IK). It used a dataset of 2,575 slit lamp images from 744 patients including normal cornea, viral keratitis (VK), fungal keratitis (FK) and bacterial keratitis (BK). They developed and tested 6 different algorithms for the classification of IK. One of those the EfficienctNetV2-M was the best even when compared with 2 ophthalmologists.
1. In Fig 1. Other than the normal cornea what was the time frame of the other photos. Are they different corneas all with the kind of keratitis stated? Better labeling or explaining is needed.
2. In Fig.6 which shows the heatmaps generated could you make clearer the text. As it stands it is not sufficient to understand what is going on. Also the use of heatmap is not what I expected, could you do a better job of explaining that.
3. Can you show the ROC curves in Fig 4 more clearly? So that their significance is better displayed?
4. It is still a problem to clearly identify BK whether by DL or by an ophthalmologist. How do you think you can improve this diagnosis? For treatment this is key.
5. You ruled out mixed infections but these are important too. How would the system be calibrated for these infections.
Author Response
Response to Reviewer 1 Comments
Thanks very much for taking your time to review this manuscript. We really appreciate all your generous comments and suggestions! We have revised the manuscript accordingly. Our point-by-point responses are detailed below.
Point 1: In Fig 1. Other than the normal cornea what was the time frame of the other photos. Are they different corneas all with the kind of keratitis stated? Better labeling or explaining is needed.
Response 1: Thanks for your suggestions. As suggested by reviewer, we have supplemented the explanation of Figure 1. (page 4)
Point 2: In Fig.6 which shows the heatmaps generated could you make clearer the text. As it stands it is not sufficient to understand what is going on. Also the use of heatmap is not what I expected, could you do a better job of explaining that.
Response 2: Thanks for your suggestions. As suggested by reviewer, we have supplemented the explanation of Figure 6. We have made a lot of attempts on heatmaps in our preliminary work, and we are so sorry they are still not that satisfying. Considering that this is a deep-learning task that needs to recognize four types of images, the main reason could be the insufficient and unbalanced amount of data we have, which prevents the model from getting comprehensive learning. We hope to improve the heatmaps attention problem in subsequent studies, and we believe that the deep learning model combined with critical manual annotation is a highly feasible approach.
Point 3: Can you show the ROC curves in Fig 4 more clearly? So that their significance is better displayed?
Response 3: Thank you for your suggestion. As suggested by reviewer, we have made some adjustments to the layout of the ROC curve images in Figure 3 and Figure 4.
Point 4: It is still a problem to clearly identify BK whether by DL or by an ophthalmologist. How do you think you can improve this diagnosis? For treatment this is key.
Response 4: Thank you for your careful review. We have added information to the discussion about the causes and solutions for the difficulties in BK diagnosis. ( page 12)
Point 5: You ruled out mixed infections but these are important too. How would the system be calibrated for these infections.
Response 5: Thank you for your careful review. We have added information to the discussion about the reasons and solutions for the difficulties in identifying mixed infections. ( page 13)

Reviewer 2 Report
I agree for the scientific principle of the work, however, the authors have to redraft the manuscript and organize the sections, tables, and figures according to the guidelines of the journal.
Tables 1 and 3 are improperly inserted.
The manuscript requires extensive editing for language, grammar, and punctuation.
The authors have to mention in their discussion the advantages of their AI based approach over other methods, please compare to these references in your discussion.
https://doi.org/10.3390/antibiotics11101374
https://doi.org/10.3390/microorganisms9061131
Author Response
Response to Reviewer 2 Comments
Thanks very much for taking your time to review this manuscript. We really appreciate all your generous comments and suggestions! We have revised the manuscript accordingly. Our point-by-point responses are detailed below.
Point 1: I agree for the scientific principle of the work, however, the authors have to redraft the manuscript and organize the sections, tables, and figures according to the guidelines of the journal.
Response 1: Thank you for your suggestion. As suggested by reviewer, we have redrafted the manuscript and organized the sections, tables, and figures according to the guidelines of the journal.
Point 2: Tables 1 and 3 are improperly inserted.
Response 2: Thank you for your suggestion. As suggested by reviewer, We have inserted the tables correctly.
Point 3: The manuscript requires extensive editing for language, grammar, and punctuation.
Response 3: We apologize for the language problems in the original manuscript. The language presentation was improved with assistance from a native English speaker with appropriate research background.
Point 4: The authors have to mention in their discussion the advantages of their AI based approach over other methods, please compare to these references in your discussion.
https://doi.org/10.3390/antibiotics11101374IF: 5.222 Q1
https://doi.org/10.3390/microorganisms9061131IF: 4.926 Q2
Response 4: Thank you for your careful review. These two important pieces of research may have found effective treatments to drug-resistant bacterial corneal infections. We cited these two articles in the discussion.

Round 2
Reviewer 1 Report
The manuscript is revised according to submitted criticisms
Reviewer 2 Report
I have no more comments, and I would accept the manuscript in its present form for publication.